# On the Convergence Rate of Training Recurrent Neural Networks*

**Zeyuan Allen-Zhu**
Microsoft Research AI
zeyuan@csail.mit.edu

**Yuanzhi Li**
Carnegie Mellon University
yuanzhil@andrew.cmu.edu

**Zhao Song**
UT-Austin
zhaos@utexas.edu

## Abstract

How can local-search methods such as stochastic gradient descent (SGD) avoid bad local minima in training multi-layer neural networks? Why can they fit random labels even given non-convex and non-smooth architectures? Most existing theory only covers networks with one hidden layer, so can we go deeper?

In this paper, we focus on recurrent neural networks (RNNs) which are multi-layer networks widely used in natural language processing. They are harder to analyze than feedforward neural networks, because the *same* recurrent unit is repeatedly applied across the entire time horizon of length $L$, which is analogous to feedforward networks of depth $L$. We show when the number of neurons is sufficiently large, meaning polynomial in the training data size and in $L$, then SGD is capable of minimizing the regression loss in the linear convergence rate. This gives theoretical evidence of how RNNs can memorize data.

More importantly, in this paper we build general toolkits to analyze multi-layer networks with ReLU activations. For instance, we prove why ReLU activations can prevent exponential gradient explosion or vanishing, and build a perturbation theory to analyze first-order approximation of multi-layer networks.

## 1 Introduction

Neural networks have been one of the most powerful tools in machine learning over the past a few decades. The multi-layer structure of neural network gives it supreme power in expressibility and learning performance. However, it raises complexity concerns: the training objective is generally *non-convex* and *non-smooth*. In practice, local-search algorithms such as stochastic gradient descent (SGD) are capable of finding global optima, at least on the training data [19, 59]. How SGD avoids local minima for such objectives remains an open theoretical question since Goodfellow et al. [19].

In recent years, there have been a number of theoretical results aiming at a better understanding of this phenomenon. Many of them focus on two-layer (thus one-hidden-layer) neural networks and assume that the inputs are random Gaussian or sufficiently close to Gaussian [7, 15, 18, 32, 37, 50, 55, 60, 61]. Some study deep neural networks but assuming the activation function is linear [5, 6, 22]. Some study the convex task of training essentially only the last layer of the network [13].

More recently, Safran and Shamir [43] provided evidence that, even when inputs are standard Gaussians, two-layer neural networks can indeed have spurious local minima, and suggested that *over-parameterization* (i.e., increasing the number of neurons) may be the key in avoiding spurious local minima. Li and Liang [31] showed that, for two-layer networks with the cross-entropy loss, in the over-parametrization regime, gradient descent (GD) is capable of finding nearly-global optimal solutions on the training data. This result was later extended to the $\ell_2$ loss by Du et al. [16].

In this paper, we show GD and SGD are capable of training multi-layer neural networks (with ReLU activation) to global minima on any non-degenerate training data set. Furthermore, the running time

is polynomial in the number of layers and the number of data points. Since there are many different types of multi-layer networks (convolutional, feedforward, recurrent, etc.), in this present paper, we focus on recurrent neural networks (RNN) as our choice of multi-layer networks, and feedforward networks are only its "special case" (see our follow-up work [3]).

**Recurrent Neural Networks.** Among different architectures of neural networks, one of the *least* theoretically-understood structure is the recurrent one [17]. A recurrent neural network recurrently applies the same network unit to a sequence of input tokens, such as a sequence of words in a language sentence. RNN is particularly useful when there are long-term, non-linear interactions between input tokens in the same sequence. These networks are widely used in practice for natural language processing, language generation, machine translation, speech recognition, video and music processing, and many other tasks [11, 12, 27, 35, 36, 44, 52, 54]. On the theory side, while there are some attempts to show that an RNN is more expressive than a feedforward neural network [28], when and how an RNN can be efficiently learned has nearly-zero theoretical explanation.

In practice, RNN is usually trained by simple local-search algorithms such as SGD. However, unlike shallow networks, the training process of RNN often runs into the trouble of vanishing or exploding gradient [53]. That is, the value of the gradient becomes exponentially small or large in the time horizon, even when the training objective is still constant. In practice, one of the popular ways to resolve this is by the long short term memory (LSTM) structure [24]. However, one can also use rectified linear units (ReLUs) as activation functions to avoid vanishing or exploding gradient [45]. In fact, one of the earliest adoptions of ReLUs was on applications of RNNs for this purpose twenty years ago [21, 46]. For a detailed survey on RNN, we refer the readers to Salehinejad et al. [45].

## 1.1 Our Question

In this paper, we study the following general question

- *Can ReLU provably stabilize the training process and avoid vanishing/exploding gradient?*
- *Can RNN be trained close to zero training error* efficiently *under mild assumptions?*

(When there is no activation function, RNN is known as *linear dynamical system* and Hardt et al. [23] proved the convergence for such linear dynamical systems.)

**Motivations.** One may also want to study whether RNN can be trained close to zero test error. However, unlike feedforward networks, the training error, or the ability to memorize examples, may actually be desirable for RNN. After all, many tasks involving RNN are related to memories, and certain RNN units are even referred to memory cells. Since RNN applies the same network unit to all input tokens in a sequence, the following question can possibly of its own interest:

- *How does RNN learn mappings (say from token 3 to token 7) without destroying others?*

Another motivation is the following. An RNN can be viewed as a space constraint, differentiable Turing machine, except that the input is only allowed to be read in a fixed order. It was shown in Siegelmann and Sontag [49] that all Turing machines can be simulated by recurrent networks built of neurons with non-linear activations. In practice, RNN is also used as a tool to build neural Turing machines [20], equipped with a grand goal of automatically learning an algorithm based on the observation of the inputs and outputs. To this extent, we believe the task of understanding the *trainability* as a first step towards understanding RNN can be meaningful on its own.

**Our Result.** To present the simplest result, we focus on the classical Elman network with ReLU activation:

$$h_\ell = \phi(W \cdot h_{\ell-1} + Ax_\ell) \in \mathbb{R}^m \qquad \text{where } W \in \mathbb{R}^{m \times m}, A \in \mathbb{R}^{m \times d_x}$$
$$y_\ell = B \cdot h_\ell \in \mathbb{R}^d \qquad \text{where } B \in \mathbb{R}^{d \times m}$$

We denote by $\phi$ the ReLU activation function: $\phi(x) = \max(x, 0)$. We note that (fully-connected) feedforward networks are only "special cases" to this by replacing $W$ with $W_\ell$ for each layer.[2]

We consider a regression task where each sequence of inputs consists of vectors $x_1, \ldots, x_L \in \mathbb{R}^{d_x}$ and we perform least-square regression with respect to $y_1^*, \ldots, y_L^* \in \mathbb{R}^d$. We assume there are $n$ training sequences, each of length $L$. We assume the training sequences are $\delta$-separable (say vectors

$x_1$ are different by relative distance $\delta > 0$ for every pairs of training sequences). Our main theorem can be stated as follows

**Theorem.** *If the number of neurons $m \geq \mathsf{poly}(n, d, L, \delta^{-1}, \log \varepsilon^{-1})$ is polynomially large, we can find weight matrices $W, A, B$ where the RNN gives $\varepsilon$ training error*

- *if gradient descent (GD) is applied for $T = \Omega\big(\frac{\mathsf{poly}(n,d,L)}{\delta^2} \log \frac{1}{\varepsilon}\big)$ iterations, starting from random Gaussian initializations; or*

- *if (mini-batch or regular) stochastic gradient descent (SGD) is applied for $T = \Omega\big(\frac{\mathsf{poly}(n,d,L)}{\delta^2} \log \frac{1}{\varepsilon}\big)$ iterations, starting from random Gaussian initializations.*[3]

**Our Contribution.** We summarize our contributions as follows.

- We believe this is the first proof of convergence of GD/SGD for training the *hidden layers* of recurrent neural networks (or even for any multi-layer networks of more than two layers) when activation functions are present.[4]

- Our results provide arguably the first theoretical evidence towards the empirical finding of Goodfellow et al. [19] on multi-layer networks, regarding the ability of SGD to avoid (spurious) local minima. Our theorem does not exclude the existence of bad local minima

- We build new technical toolkits to analyze multi-layer networks with ReLU activation, which have now found many applications [1–3, 9]. For instance, combining this paper with new techniques, one can derive guarantees on testing error for RNN in the PAC-learning language [1].

**Extension: DNN.** A feedforward neural network of depth $L$ is similar to Elman RNN with the main difference being that the weights across layers are separately trained. As one shall see, this only makes our proofs *simpler* because we have more independence in randomness. Our theorems also apply to feedforward neural networks, and we have written a separate follow-up paper [3] to address feedforward (fully-connected, residual, and convolutional) neural networks.

EXTENSION: DEEP RNN. Elman RNN is also referred to as three-layer RNN, and one may also study the convergence of RNNs with more hidden layers. This is referred to as deep RNN [45]. Our theorem also applies to deep RNNs (by combining this paper together with [3]).

EXTENSION: LOSS FUNCTIONS. For simplicity, in this paper we have adopted the $\ell_2$ regression loss. Our results generalize to other Lipschitz smooth (but possibly nonconvex) loss functions, by combining with the techniques of [3].

## 1.2 Other Related Works

Another relevant work is Brutzkus et al. [8] where the authors studied over-paramterization in the case of two-layer neural network under a linear-separable assumption.

Instead of using randomly initialized weights like this paper, there is a line of work proposing algorithms using weights generated from some "tensor initialization" process [4, 26, 48, 55, 61].

There is huge literature on using the mean-field theory to study neural networks [10, 14, 25, 30, 34, 38–40, 42, 47, 56–58]. At a high level, they study the network dynamics at random initialization when the number of hidden neurons grow to infinity, and use such initialization theory to predict performance after training. However, they do not provide theoretical convergence rate for the training process (at least when the number of neurons is finite).

# 2 Notations and Preliminaries

We denote by $\|\cdot\|_2$ (or sometimes $\|\cdot\|$) the Euclidean norm of vectors, and by $\|\cdot\|_2$ the spectral norm of matrices. We denote by $\|\cdot\|_\infty$ the infinite norm of vectors, $\|\cdot\|_0$ the sparsity of vectors or diagonal matrices, and $\|\cdot\|_F$ the Frobenius norm of matrices. Given matrix $W$, we denote by $W_k$ or $w_k$ the $k$-th row vector of $W$. We denote the row $\ell_p$ norm for $W \in \mathbb{R}^{m \times d}$ as $\|W\|_{2,p} := \left( \sum_{i \in [m]} \|w_i\|_2^p \right)^{1/p}$. By definition, $\|W\|_{2,2} = \|W\|_F$.

We use $\mathcal{N}(\mu, \sigma)$ to denote Gaussian distribution with mean $\mu$ and variance $\sigma$; or $\mathcal{N}(\mu, \Sigma)$ to denote Gaussian vector with mean $\mu$ and covariance $\Sigma$. We use $\mathbb{1}_{event}$ to denote the indicator function of whether $event$ is true. We denote by $\mathbf{e}_k$ the $k$-th standard basis vector. We use $\phi(\cdot)$ to denote the ReLU function, namely $\phi(x) = \max\{x, 0\} = \mathbb{1}_{x \geq 0} \cdot x$. Given univariate function $f \colon \mathbb{R} \to \mathbb{R}$, we also use $f$ to denote the same function over vectors: $f(x) = (f(x_1), \ldots, f(x_m))$ if $x \in \mathbb{R}^m$.

Given vectors $v_1, \ldots, v_n \in \mathbb{R}^m$, we define $U = \mathsf{GS}(v_1, \ldots, v_n)$ as their Gram-Schmidt orthonormalization. Namely, $U = [\widehat{v}_1, \ldots, \widehat{v}_n] \in \mathbb{R}^{m \times n}$ where

$$\widehat{v}_1 = \frac{v_1}{\|v_1\|} \quad \text{and} \quad \text{for } i \geq 2 \colon \quad \widehat{v}_i = \frac{\prod_{j=1}^{i-1}(I - \widehat{v}_j \widehat{v}_j^\top) v_i}{\|\prod_{j=1}^{i-1}(I - \widehat{v}_j \widehat{v}_j^\top) v_i\|}.$$

Note that in the occasion that $\prod_{j=1}^{i-1}(I - \widehat{v}_j \widehat{v}_j^\top) v_i$ is the zero vector, we let $\widehat{v}_i$ be an arbitrary unit vector that is orthogonal to $\widehat{v}_1, \ldots, \widehat{v}_{i-1}$.

## 2.1 Elman Recurrent Neural Network

We assume $n$ training inputs are given: $(x_{i,1}, x_{i,2}, \ldots, x_{i,L}) \in \left(\mathbb{R}^{d_x}\right)^L$ for each input $i \in [n]$. We assume $n$ training labels are given: $(y_{i,1}^*, y_{i,2}^*, \ldots, y_{i,L}^*) \in \left(\mathbb{R}^d\right)^L$ for each input $i \in [n]$. Without loss of generality, we assume $\|x_{i,\ell}\| \leq 1$ for every $i \in [n]$ and $\ell \in [L]$. Also without loss of generality, we assume $\|x_{i,1}\| = 1$ and its last coordinate $[x_{i,1}]_{d_x} = \frac{1}{\sqrt{2}}$ for every $i \in [n]$.[5]

We make the following assumption on the input data (see Footnote 9 for how to relax it):

**Assumption 2.1.** $\|x_{i,1} - x_{j,1}\| \geq \delta$ *for some parameter* $\delta \in (0, 1]$ *and every pair of* $i \neq j \in [n]$.

Given weight matrices $W \in \mathbb{R}^{m \times m}$, $A \in \mathbb{R}^{m \times d_x}$, $B \in \mathbb{R}^{d \times m}$, we introduce the following notations to describe the evaluation of RNN on the input sequences. For each $i \in [n]$ and $j \in [L]$:

$$h_{i,0} = 0 \in \mathbb{R}^m \qquad\qquad g_{i,\ell} = W \cdot h_{i,\ell-1} + A x_{i,\ell} \in \mathbb{R}^m$$
$$y_{i,\ell} = B \cdot h_{i,\ell} \in \mathbb{R}^d \qquad\qquad h_{i,\ell} = \phi(W \cdot h_{i,\ell-1} + A x_{i,\ell}) \in \mathbb{R}^m$$

A very important notion that this entire paper relies on is the following:

**Definition 2.2.** *For each* $i \in [n]$ *and* $\ell \in [L]$, *let* $D_{i,\ell} \in \mathbb{R}^{m \times m}$ *be the diagonal matrix where*

$$(D_{i,\ell})_{k,k} = \mathbb{1}_{(W \cdot h_{i,\ell-1} + A x_{i,\ell})_k \geq 0} = \mathbb{1}_{(g_{i,\ell})_k \geq 0} \ .$$

*As a result, we can write* $h_{i,\ell} = D_{i,\ell} W h_{i,\ell-1}$.

We consider the following random initialization distributions for $W$, $A$ and $B$.

**Definition 2.3.** *We say that* $W, A, B$ *are at random initialization, if the entries of* $W$ *and* $A$ *are i.i.d. generated from* $\mathcal{N}(0, \frac{2}{m})$, *and the entries of* $B_{i,j}$ *are i.i.d. generated from* $\mathcal{N}(0, \frac{1}{d})$.

Throughout this paper, for notational simplicity, we refer to index $\ell$ as the $\ell$-th *layer* of RNN, and $h_{i,\ell}, x_{i,\ell}, y_{i,\ell}$ respectively as the hidden neurons, input, output on the $\ell$-th layer. We acknowledge that in certain literatures, one may regard Elman network as a three-layer RNN.

**Assumption 2.4.** *We assume* $m \geq \mathsf{poly}(n, d, L, \frac{1}{\delta}, \log \frac{1}{\varepsilon})$ *for some sufficiently large polynomial.*

Without loss of generality, we assume $\delta \leq \frac{1}{CL^2 \log^3 m}$ for some sufficiently large constant $C$ (if this is not satisfied one can decrease $\delta$). Throughout the paper except the detailed appendix, we use $\widetilde{O}$, $\widetilde{\Omega}$ and $\widetilde{\Theta}$ notions to hide polylogarithmic dependency in $m$. To simplify notations, we denote by

$$\rho := nLd \log m \quad \text{and} \quad \varrho := nLd\delta^{-1} \log(m/\varepsilon) \ .$$

## 2.2 Objective and Gradient

For simplicity, we only optimize over the weight matrix $W \in \mathbb{R}^{m \times m}$ and let $A$ and $B$ be at random initialization. As a result, our $\ell_2$-regression objective is a function over $W$:[6]

$$f(W) := \sum_{i=1}^{n} f_i(W) \quad \text{and} \quad f_i(W) := \frac{1}{2} \sum_{\ell=2}^{L} \| \text{loss}_{i,\ell} \|_2^2 \quad \text{where} \quad \text{loss}_{i,\ell} := Bh_{i,\ell} - y_{i,\ell}^* .$$

Using chain rule, one can write down a closed form of the (sub-)gradient:

**Fact 2.5.** *For $k \in [m]$, the gradient with respect to $W_k$ (denoted by $\nabla_k$) and the full gradient are*

$$\nabla_k f(W) = \sum_{i=1}^{n} \sum_{a=2}^{L} \sum_{\ell=1}^{a-1} (\text{Back}_{i,\ell+1 \to a}^{\top} \cdot \text{loss}_{i,a})_k \cdot h_{i,\ell} \cdot \mathbf{1}_{\langle W_k, h_{i,\ell} \rangle + \langle A_k, x_{i,\ell+1} \rangle \geq 0}$$

$$\nabla f(W) = \sum_{i=1}^{n} \sum_{a=2}^{L} \sum_{\ell=1}^{a-1} D_{i,\ell+1} (\text{Back}_{i,\ell+1 \to a}^{\top} \cdot \text{loss}_{i,a}) \cdot h_{i,\ell}^{\top}$$

*where for every $i \in [n]$, $\ell \in [L]$, and $a = \ell + 1, \ell + 2, \dots, L$:*

$$\text{Back}_{i,\ell \to \ell} := B \in \mathbb{R}^{d \times m} \quad \text{and} \quad \text{Back}_{i,\ell \to a} := BD_{i,a}W \cdots D_{i,\ell+1}W \in \mathbb{R}^{d \times m} .$$

# 3 Our Results

Our main results can be formally stated as follows.

**Theorem 1** (GD). *Suppose $\eta = \widetilde{\Theta}\big(\frac{\delta}{m}\text{poly}(n, d, L)\big)$ and $m \geq \text{poly}(n, d, L, \delta^{-1}, \log \varepsilon^{-1})$. Let $W^{(0)}, A, B$ be at random initialization. With high probability over the randomness of $W^{(0)}, A, B$, if we apply gradient descent for $T$ steps $W^{(t+1)} = W^{(t)} - \eta \nabla f(W^{(t)})$, then it satisfies*

$$f(W^{(T)}) \leq \varepsilon \quad \text{for} \quad T = \widetilde{\Omega}\big(\frac{\text{poly}(n, d, L)}{\delta^2} \log \frac{1}{\varepsilon}\big).$$

**Theorem 2** (SGD). *Suppose $\eta = \widetilde{\Theta}\big(\frac{\delta}{m}\text{poly}(n, d, L)\big)$ and $m \geq \text{poly}(n, d, L, \delta^{-1}, \log \varepsilon^{-1})$. Let $W^{(0)}, A, B$ be at random initialization. If we apply stochastic gradient descent for $T$ steps $W^{(t+1)} = W^{(t)} - \eta \nabla f_i(W^{(t)})$ for a random index $i \in [n]$ per step, then with high probability (over $W^{(0)}, A, B$ and the randomness of SGD), it satisfies*

$$f(W^{(T)}) \leq \varepsilon \quad \text{for} \quad T = \widetilde{\Omega}\big(\frac{\text{poly}(n, d, L)}{\delta^2} \log \frac{1}{\varepsilon}\big).$$

In both cases, we essentially have *linear convergence rates*. Notably, our results show that the dependency of the number of layers $L$, is *polynomial*. Thus, even when RNN is applied to sequences of long input data, it does not suffer from exponential gradient explosion or vanishing (e.g., $2^{\Omega(L)}$ or $2^{-\Omega(L)}$) through the entire training process.

**Main Technical Theorems.** Our main Theorem 1 and Theorem 2 are in fact natural consequences of the following two technical theorems. They both talk about the first-order behavior of RNNs when the weight matrix $W$ is sufficiently close to some random initialization.

The first theorem is similar to the classical Polyak-Łojasiewicz condition [33, 41], and says that $\|\nabla f(W)\|_F^2$ is at least as large as the objective value.

**Theorem 3.** *With high probability over random initialization $\widetilde{W}, A, B$, it satisfies*

$$\forall W \in \mathbb{R}^{m \times m} \text{ with } \|W - \widetilde{W}\|_2 \leq \frac{\text{poly}(\varrho)}{\sqrt{m}} : \qquad \|\nabla f(W)\|_F^2 \geq \frac{\delta}{\text{poly}(\rho)} \times m \times f(W) ,$$

$$\|\nabla f(W)\|_F^2, \|\nabla f_i(W)\|_F^2 \leq \text{poly}(\rho) \times m \times f(W) .$$

(Only the first statement is the Polyak-Łojasiewicz condition; the second is a simple-to-proof gradient upper bound.) The second theorem shows a special "semi-smoothness" property of the objective.

**Theorem 4.** *With high probability over random initialization $\widetilde{W}, A, B$, it satisfies for every $\breve{W} \in \mathbb{R}^{m \times m}$ with $\|\breve{W} - \widetilde{W}\| \leq \frac{\text{poly}(\varrho)}{\sqrt{m}}$, and for every $W' \in \mathbb{R}^{m \times m}$ with $\|W'\| \leq \frac{\tau_0}{\sqrt{m}}$,*

$$f(\breve{W} + W') \leq f(\breve{W}) + \langle \nabla f(\breve{W}), W' \rangle + \text{poly}(\varrho)m^{1/3} \cdot \sqrt{f(W)} \cdot \|W'\|_2 + \text{poly}(\rho)m\|W'\|_2^2 .$$

At a high level, the convergence of GD and SGD are careful applications of the two technical theorems above: indeed, Theorem 3 shows that as long as the objective value is high, the gradient is large; and Theorem 4 shows that if one moves in the (negative) gradient direction, then the objective value can be sufficiently decreased. These two technical theorems together ensure that GD/SGD does not hit any saddle point or (bad) local minima along its training trajectory. This was practically observed by Goodfellow et al. [19] and a theoretical justification was open since then.

**An Open Question.** We did not try to tighten the polynomial dependencies of $(n, d, L)$ in the proofs. When $m$ is sufficiently large, we make use of the randomness at initialization to argue that, *for all the points* within a certain radius from initialization, for instance Theorem 3 holds. In practice, however, the SGD can create additional randomness as time goes; also, in practice, it suffices for those points on the SGD trajectory to satisfy Theorem 3. Unfortunately, such randomness can — in principle — be correlated with the SGD trajectory, so we do not know how to use that in the proofs. Analyzing such correlated randomness is certainly beyond the scope of this paper, but can possibly explain why in practice, the size of $m$ needed is not that large.

## 3.1 Conclusion

Overall, we provide the first proof of convergence of GD/SGD for non-linear neural networks that have more two layers. We show with overparameterization GD/SGD can avoid hitting any (bad) local minima along its training trajectory. This was practically observed by Goodfellow et al. [19] and a theoretical justification was open since then. We present our result using recurrent neural networks (as opposed to the simpler feedforward networks [3]) in this very first paper, because memorization in RNN could be of independent interest. Also, our result proves that RNN can learn mappings from different input tokens to different output tokens *simultaneously* using the same recurrent unit.

Last but not least, we build new tools to analyze multi-layer networks with ReLU activations that could facilitate many new research on deep learning. For instance, our techniques in Section 4 provide a general theory for why ReLU activations avoid exponential exploding (see e.g. (4.1), (4.4)) or exponential vanishing (see e.g. (4.1), (4.3)); and our techniques in Section 5 give a general theory for the stability of multi-layer networks against adversarial weight perturbations, which is at the heart of showing the semi-smoothness Theorem 4, and used by all the follow-up works [1–3, 9].

# PROOF SKETCH

The main difficulty of this paper is to prove Theorem 3 and 4, and we shall sketch the proof ideas in Section 4 through 8. In this main body, we only include Section 4 and 5 because they already given some insights into how the proof proceeds. We shall put our emphasize on

- how to avoid exponential blow up in $L$, and
- how to deal with the issue of randomness dependence across layers.

We genuinely hope that this high-level sketch can (1) give readers a clear overview of the proof without the necessity of going to the appendix, and (2) appreciate our proof and understand why it is necessarily long.[7]

## 4 Basic Properties at Random Initialization

In this section we derive basic properties of the RNN when the weight matrices $W, A, B$ are all *at random initialization*. The corresponding precise statements and proofs are in Appendix B.

The first one says that the forward propagation neither explodes or vanishes, that is,

$$\frac{1}{2} \le \|h_{i,\ell}\|_2, \|g_{i,\ell}\|_2 \le O(L) \ . \tag{4.1}$$

Intuitively, (4.1) very reasonable. Since the weight matrix $W$ is randomly initialized with entries i.i.d. from $\mathcal{N}\left(0, \frac{2}{m}\right)$, the norm $\|Wz\|_2$ is around $\sqrt{2}$ for any fixed vector $z$. Equipped with ReLU activation, it "shuts down" roughly half of the coordinates of $Wz$ and reduces the norm $\|\phi(Wz)\|$ to one. Since in each layer $\ell$, there is an additional unit-norm signal $x_{i,\ell}$ coming in, we should expect the final norm of hidden neurons to be at most $O(L)$.

Unfortunately, the above argument cannot be directly applied since the weight matrix $W$ is reused for $L$ times so there is no fresh new randomness across layers. Let us explain how we deal with this issue carefully, because it is at the heart of *all* of our proofs in this paper. Recall, each time $W$ is applied to some vector $h_{i,\ell}$, it only uses "one column of randomness" of $W$. Mathematically, letting $U_\ell \in \mathbb{R}^{m \times n\ell}$ denote the column orthonormal matrix using Gram-Schmidt

$$U_\ell := \mathsf{GS}\left(h_{1,1}, \ldots, h_{n,1}, h_{1,2}, \ldots, h_{n,2}, \ldots, h_{1,\ell}, \ldots, h_{n,\ell}\right) \quad,$$

we have $Wh_{i,\ell} = WU_{\ell-1}U_{\ell-1}^\top h_{i,\ell} + W(I - U_{\ell-1}U_{\ell-1}^\top)h_{i,\ell}$.

- The term $W(I - U_{\ell-1}U_{\ell-1}^\top)h_{i,\ell}$ has new randomness independent of the previous layers.[8]
- The term $WU_{\ell-1}U_{\ell-1}^\top h_{i,\ell}$ relies on the randomness of $W$ in the directions of $h_{i,a}$ for $a < \ell$ of the previous layers. We cannot rely on the randomness of this term, because when applying inductive argument till layer $\ell$, the randomness of $WU_{\ell-1}$ is already used.

  Fortunately, $WU_{\ell-1} \in \mathbb{R}^{m \times n(\ell-1)}$ is a rectangular matrix with $m \gg n(\ell - 1)$ (thanks to overparameterization!) so one can bound its spectral norm by roughly $\sqrt{2}$. This ensures that no matter how $h_{i,\ell}$ behaves (even arbitrarily correlated with $WU_{\ell-1}$), the norm of the first term cannot be too large. It is crucial here that $WU_{\ell-1}$ is a *rectangular* matrix, because for a square random matrix such as $W$, its spectral norm is 2 and using that, the forward propagation bound will exponentially blow up.

This summarizes the main idea for proving $\|h_{i,\ell}\| \leq O(L)$ in (4.1); the lower bound $\frac{1}{2}$ is similar. Our next property says in each layer, the amount of "fresh new randomness" is non-negligible:

$$\|(I - U_{\ell-1}U_{\ell-1}^\top)h_{i,\ell}\|_2 \geq \widetilde{\Omega}(\frac{1}{L^2}) \quad. \tag{4.2}$$

This relies on a more involved inductive argument than (4.1). At high level, one needs to show that in each layer, the amount of "fresh new randomness" reduces only by a factor at most $1 - \frac{1}{10L}$.

Using (4.1) and (4.2), we obtain the following property about the data separability:

$$(I - U_\ell U_\ell^\top)h_{i,\ell+1} \text{ and } (I - U_\ell U_\ell^\top)h_{j,\ell+1} \text{ are } (\delta/2)\text{-separable}, \forall i, j \in [n] \text{ with } i \neq j \tag{4.3}$$

Here, we say two vectors $x$ and $y$ are $\delta$-separable if $\left\|(I - yy^\top/\|y\|_2^2)x\right\| \geq \delta$ and vice versa. Property (4.3) shows that the separability information (say on input token 1) *does not diminish* by more than a polynomial factor even if the information is propagated for $L$ layers.

We prove (4.3) by induction. In the first layer $\ell = 1$ we have $h_{i,1}$ and $h_{j,1}$ are $\delta$-separable which is a consequence of Assumption 2.1. If having fresh new randomness, given two $\delta$ separable vectors $x, y$, one can show that $\phi(Wx)$ and $\phi(Wy)$ are also $\delta(1 - o(\frac{1}{L}))$-separable. Again, in RNN, we do not have fresh new randomness, so we rely on (4.2) to give us reasonably large fresh new randomness. Applying a careful induction helps us to derive that (4.3) holds for all layers.[9]

**Intermediate Layers and Backward Propagation.** Training neural network is not only about forward propagation. We also have to bound intermediate layers and backward propagation.

The first two results we derive are the following. For every $\ell_1 \geq \ell_2$ and diagonal matrices $D'$ of sparsity $s \in [\rho^2, m^{0.49}]$:

$$\|WD_{i,\ell_1} \cdots D_{i,\ell_2}W\|_2 \leq O(L^3) \tag{4.4}$$

$$\|D'WD_{i,\ell_1} \cdots D_{i,\ell_2}WD'\|_2 \leq \widetilde{O}(\sqrt{s}/\sqrt{m}) \tag{4.5}$$

Intuitively, one cannot use spectral bound argument to derive (4.4) or (4.5): the spectral norm of $W$ is 2, and even if ReLU activations cancel half of its mass, the spectral norm $\|DW\|_2$ remains to be $\sqrt{2}$. When stacked together, this grows exponential in $L$.

Instead, we use an analogous argument to (4.1) to show that, for each fixed vector $z$, the norm of $\|WD_{i,\ell_1} \cdots D_{i,\ell_2} Wz\|_2$ is at most $O(1)$ with extremely high probability $1 - e^{-\Omega(m/L^2)}$. By standard $\varepsilon$-net argument, $\|WD_{i,\ell_1} \cdots D_{i,\ell_2} Wz\|_2$ is at most $O(1)$ for all $\frac{m}{L^3}$-sparse vectors $z$. Finally, for a possible dense vector $z$, we can divide it into $L^3$ chunks each of sparsity $\frac{m}{L^3}$. Finally, we apply the upper bound for $L^3$ times. This proves (4.4). One can use similar argument to prove (4.5).

*Remark* 4.1. We did not try to tighten the polynomial factor here in $L$. We conjecture that proving an $O(1)$ bound may be possible, but that question itself may be a sufficiently interesting random matrix theory problem on its own.

The next result is for back propagation. For every $\ell_1 \geq \ell_2$ and diagonal matrices $D'$ of sparsity $s \in [\rho^2, m^{0.49}]$:

$$\|BD_{i,\ell_1} \cdots D_{i,\ell_2} WD'\|_2 \leq \widetilde{O}(\sqrt{s}) \tag{4.6}$$

Its proof is in the same spirit as (4.5), with the only difference being the spectral norm of $B$ is around $\sqrt{m/d}$ as opposed to $O(1)$.

# 5 Stability After Adversarial Perturbation

In this section we study the behavior of RNN after adversarial perturbation. The corresponding precise statements and proofs are in Appendix C.

Letting $\widetilde{W}, A, B$ be at random initialization, we consider some matrix $W = \widetilde{W} + W'$ for $\|W'\|_2 \leq \frac{\text{poly}(\varrho)}{\sqrt{m}}$. Here, $W'$ may depend on the randomness of $\widetilde{W}, A$ and $B$, so we say it can be *adversarially* chosen. The results of this section will later be applied essentially twice:

- Once for those updates generated by GD or SGD, where $W'$ is how much the algorithm has moved away from the random initialization.
- The other time (see Section 7.3) for a technique that we call "randomness decomposition" where we decompose the true random initialization $W$ into $W = \widetilde{W} + W'$, where $\widetilde{W}$ is a "fake" random initialization but identically distributed as $W$. Such technique comes from smooth analysis [51].

To illustrate our high-level idea, from this section on (so in Section 5, 7 and 8)

> we ignore the polynomial dependency in $\varrho$ and *hide it in the big-O notion*.

We denote by $\widetilde{D}_{i,\ell}, \widetilde{g}_{i,\ell}, \widetilde{h}_{i,\ell}$ respectively the values of $D_{i,\ell}, g_{i,\ell}$ and $h_{i,\ell}$ determined by $\widetilde{W}$ and $A$ at random initialization; and by $D_{i,\ell} = \widetilde{D}_{i,\ell} + D'_{i,\ell}$, $g_{i,\ell} = \widetilde{g}_{i,\ell} + g'_{i,\ell}$ and $h_{i,\ell} = \widetilde{h}_{i,\ell} + h'_{i,\ell}$ respectively those determined by $W = \widetilde{W} + W'$ after the adversarial perturbation.

**Forward Stability.** Our first, and most technical result is the following:

$$\|g'_{i,\ell}\|_2, \|h'_{i,\ell}\|_2 \leq O(m^{-1/2}) \ , \quad \|D'_{i,\ell}\|_0 \leq O(m^{2/3}) \quad \text{and} \quad \|D'_{i,\ell}g_{i,\ell}\|_2 \leq O(m^{-1/2}) \ . \tag{5.1}$$

Intuitively, one may hope to prove (5.1) by induction, because we have (ignoring subscripts in $i$)

$$g'_{\ell'} = \underbrace{W'D_{\ell'-1}g_{\ell'-1}}_{\text{①}} + \underbrace{\widetilde{W}D'_{\ell'-1}g_{\ell'-1}}_{\text{②}} + \underbrace{\widetilde{W}\widetilde{D}_{\ell'-1}g'_{\ell'-1}}_{\text{③}} \ .$$

The main issue here is that, the spectral norm of $\widetilde{W}\widetilde{D}_{\ell'-1}$ in ③ is greater than 1, so we cannot apply naive induction due to exponential blow up in $L$. Neither can we apply techniques from Section 4, because the changes such as $g_{\ell'-1}$ can be *adversarial*.

In our actual proof of (5.1), instead of applying induction on ③, we recursively expand ③ by the above formula. This results in a total of $L$ terms of ① type and $L$ terms of ② type. The main difficulty is to bound a term of ② type, that is:

$$\left\|\widetilde{W}\widetilde{D}_{\ell_1} \cdots \widetilde{D}_{\ell_2+1}\widetilde{W}D'_{\ell_2}g_{\ell_2}\right\|_2$$

Our argument consists of two conceptual steps.

(1) Suppose $g_{\ell_2} = \widetilde{g}_{\ell_2} + g'_{\ell_2} = \widetilde{g}_{\ell_2} + g'_{\ell_2,1} + g'_{\ell_2,2}$ where $\|g'_{\ell_2,1}\|_2 \leq m^{-1/2}$ and $\|g'_{\ell_2,2}\|_\infty \leq m^{-1}$, then we argue that $\|D'_{\ell_2} g_{\ell_2}\|_2 \leq O(m^{-1/2})$ and $\|D'_{\ell_2} g_{\ell_2}\|_0 \leq O(m^{2/3})$.

(2) Suppose $x \in \mathbb{R}^m$ with $\|x\|_2 \leq m^{-1/2}$ and $\|x\|_0 \leq m^{2/3}$, then we show that $y = \widetilde{W} \widetilde{D}_{\ell_1} \cdots \widetilde{D}_{\ell_2+1} \widetilde{W} x$ can be written as $y = y_1 + y_2$ with $\|y_1\|_2 \leq O(m^{-2/3})$ and $\|y_2\|_\infty \leq O(m^{-1})$.

The two steps above enable us to perform induction without exponential blow up. Indeed, they together enable us to go through the following logic chain:

$$\left.\begin{array}{c} \|\cdot\|_2 \leq m^{-1/2} \text{ and } \|\cdot\|_\infty \leq m^{-1} \quad \overset{(1)}{\Longrightarrow} \\ \|\cdot\|_2 \leq m^{-2/3} \text{ and } \|\cdot\|_\infty \leq m^{-1} \quad \underset{(2)}{\Longleftarrow} \end{array}\right\} \|\cdot\|_2 \leq m^{-1/2} \text{ and } \|\cdot\|_0 \leq m^{2/3}$$

Since there is a gap between $m^{-1/2}$ and $m^{-2/3}$, we can make sure that all blow-up factors are absorbed into this gap, using the property that $m$ is polynomially large. This enables us to perform induction to prove (5.1) without exponential blow-up.

**Intermediate Layers and Backward Stability.** Using (5.1), and especially using the sparsity $\|D'\|_0 \leq m^{2/3}$ from (5.1), one can apply the results in Section 4 to derive the following stability bounds for intermediate layers and backward propagation:

$$\left\| D_{i,\ell_1} W \cdots D_{i,\ell_2} W - \widetilde{D}_{i,\ell_1} \widetilde{W} \cdots \widetilde{D}_{i,\ell_2} \widetilde{W} \right\|_2 \leq O(L^7) \tag{5.2}$$

$$\left\| B D_{i,\ell_1} W \cdots D_{i,\ell_2} W - B \widetilde{D}_{i,\ell_1} \widetilde{W} \cdots \widetilde{D}_{i,\ell_2} \widetilde{W} \right\|_2 \leq O(m^{1/3}) . \tag{5.3}$$

**Special Rank-1 Perturbation.** For technical reasons, we also need two bounds in the special case of $W' = yz^\top$ for some unit vector $z$ and sparse $y$ with $\|y\|_0 \leq \mathsf{poly}(\varrho)$. We prove that, for this type of rank-one adversarial perturbation, it satisfies for every $k \in [m]$:

$$|((\widetilde{W} + W')h'_{i,\ell})_k| \leq O(m^{-2/3}) \tag{5.4}$$

$$\left\| B D_{i,\ell_1} W \cdots D_{i,\ell_2} W \mathbf{e}_k - B \widetilde{D}_{i,\ell_1} \widetilde{W} \cdots \widetilde{D}_{i,\ell_2} \widetilde{W} \mathbf{e}_k \right\|_2 \leq O(m^{-1/6}) \tag{5.5}$$

# 6 Conclusion and What's After Page 8

We conclude the paper here because Section 4 and 5 have already given some insights into how the proof proceeds and how to avoid exponential blow up in $L$. In the supplementary material, within another 3 pages we also sketch the proof ideas for Theorem 3 and 4 (see Section 7 and 8). We genuinely hope that this high-level sketch can (1) give readers a clear overview of the proof without the necessity of going to the appendix, and (2) appreciate our proof and understand why it is necessarily long.[10] /

Overall, we provide the first proof of convergence of GD/SGD for non-linear neural networks that have more two layers. We show with overparameterization GD/SGD can avoid hitting any (bad) local minima along its training trajectory. This was practically observed by Goodfellow et al. [19] and a theoretical justification was open since then. We present our result using recurrent neural networks (as opposed to the simpler feedforward networks [3]) in this very first paper, because memorization in RNN could be of its own independent interest. Also, our result proves that RNN can indeed learn mappings from different input tokens to different input tokens *simultaneously*.

Last but not least, we build new tools to analyze multi-layer networks with ReLU activations that could facilitate many new research on deep learning. For instance, our techniques in Section 4 provide a general theory for why ReLU activations avoid exponential exploding (see e.g. (4.1), (4.4)) or exponential vanishing (see e.g. (4.1), (4.3)); and our techniques in Section 5 give a general theory for the stability of multi-layer networks against adversarial weight perturbations, which is at the heart of showing the semi-smoothness Theorem 4, and used by all the follow-up works [1–3, 9].

## Footnotes

*Full version and future updates can be found on `https://arxiv.org/abs/1810.12065`.

[2]Most of the technical lemmas of this paper remain to hold (and become much simpler) once $W$ is replaced with $W_\ell$. This is carefully treated by [3].

[3]At a first glance, one may question how it is possible for SGD to enjoy a logarithmic time dependency in $\varepsilon^{-1}$; after all, even when minimizing strongly-convex and Lipschitz-smooth functions, the typical convergence rate of SGD is $T \propto 1/\varepsilon$ as opposed to $T \propto \log(1/\varepsilon)$. We quickly point out there is no contradiction here if the stochastic pieces of the objective enjoy a *common* global minimizer.

[4]Our theorem holds even when $A, B$ are at random initialization and only the hidden weight matrix $W$ is trained. This is much more difficult to analyze than the convex task of training only the last layer $B$ [13]. Training only the last layer can significantly reduce the learning power of (recurrent or not) neural networks in practice.

[5]If it only satisfies $\|x_{i,1}\| \leq 1$ one can pad it with an additional coordinate to make $\|x_{i,1}\| = 1$ hold. As for the assumption $[x_{i,1}]_{d_x} = \frac{1}{\sqrt{2}}$, this is equivalent to adding a bias term $\mathcal{N}(0, \frac{1}{m})$ for the first layer.

[6]The index $\ell$ starts from 2, because $Bh_{i,1} = B\phi(Ax_{i,1})$ remains constant if we are not optimizing over $A$ and $B$.

[7]For instance, proving gradient norm lower bound in Theorem 3 for a single neuron $k \in [m]$ is easy, but how to apply concentration across neurons? Crucially, due to the recurrent structure these quantities are never independent, so we have to build necessary probabilistic tools to tackle this. If one is willing to ignore such subtleties, then our sketched proof is sufficiently short and gives a good overview.

[8]More precisely, letting $v = (I - U_{\ell-1}U_{\ell-1}^\top)h_{i,\ell}$, we have $W(I - U_{\ell-1}U_{\ell-1}^\top)h_{i,\ell} = \left(W\frac{v}{\|v\|}\right)\|v\|$. Here, $W\frac{v}{\|v\|}$ is a random Gaussian vector in $\mathcal{N}(0, \frac{2}{m}I)$ and is *independent* of all $\{h_{i,a} \mid i \in [n], a < \ell\}$.

[9]This is the only place that we rely on Assumption 2.1. This assumption is somewhat necessary in the following sense. If $x_{i,\ell} = x_{j,\ell}$ for some pair $i \neq j$ for all the first ten layers $\ell = 1, 2, \ldots, 10$, and if $y_{i,\ell}^* \neq y_{i,\ell}^*$ for even just one of these layers, then there is no hope in having the training objective decrease to zero. Of course, one can make more relaxed assumption on the input data, involving both $x_{i,\ell}$ and $y_{i,\ell}^*$. While this is possible, it complicates the statements so we do not present such results in this paper.

[10] For instance, proving gradient norm lower bound in Theorem 3 for a single neuron $k \in [m]$ is easy, but how to apply concentration across neurons? Crucially, due to the recurrent structure these quantities are never independent, so we have to build necessary probabilistic tools to tackle this. If one is willing to ignore such subtleties, then our sketched proof is sufficiently short and gives a good overview.

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
