[Reviews · NeurIPS 2019]

Reviewer 1



This paper shows that GD/SGD can minimize the training loss of RNNs with linear convergence rate assuming the hidden layer width is sufficiently large (polynomial in data size and time horizon length). In order to prove this, the authors show that within a small region around the initialization, the norm square of the gradient can be lower bounded by the function value (Theorem 3). The authors further show that the loss function is somewhat smooth (Theorem 4), which guarantees that moving in the negative gradient direction can decrease the function value. This paper builds new techniques to analyze multi-layer ReLU networks. This paper shows that with appropriate initialization, ReLU activations avoid exponential exploding and exponential vanishing. This paper also shows that within a small region around the initialization, the multi-layer networks is pretty smooth’’. These techniques are very useful in the analysis of multi-layer networks and have been used in many following works. This paper is an important step towards the optimization theory of RNNs. The RNNs is much harder to analyze because the same recurrent unit is repeated applied and at initialization the randomness is shared across layers. This paper uses randomness decoupling techniques to analyze the spectral norm of RNNs at initialization. These techniques can be useful in other problems of RNNs. Overall this is a strong theory paper proving that GD/SGD can optimize RNNs in the over-parameterized setting. The techniques developed in this paper can be very useful in the analysis of multi-layer ReLU nets. My only concern is that the required hidden layer width is a polynomial in the number of samples (might be high order polynomial), which is not very practical. Also, the step size is very small, and the total movement of the weights is very small. In practice, the step size is much larger, and the weights move a lot. So, in some sense, this theory cannot explain the success of over-parameterization in practice. Reducing the dependency on m might require a very different idea. Here are some minor comments: 1. Line 142: shouldn’t it be h = D(Wh + Ax)? Ax is missing here. 2. It might be good to add a figure to illustrate the network architecture if space allows. -------------------------------------------- I have read the authors' response and other reviews. The authors have partially addressed my concerns on the network width and step size. I agree that as long as we assume the data is generated by some simple model, the requirement on the network width can be significantly reduced. Regarding the step size, the authors argue that this work can give some intuitions on the second phase of NN training when the step size decays and training loss goes to zero. However, the weights have moved a lot in the first phase (when the step size is large) and are not random anymore, it's not clear whether the current techniques can still work after phase 1. Despite these limitations, I still think this is a good theory paper and I will keep my score.

Reviewer 2



The paper proves that with overparameterized RNNs (polynomial in the training data), GD/SGD can find a global minima in polynomial time, with high probability, if the neural network is initialized according to some distribution (and is based on a three-layer RNN with ReLU activation functions). The paper does not eliminate the existence of worse local minima for proving these results, which means even if there are bad local minima, with high probability GD/SGD can still find a global minima. This could explain why in practice overparameterized networks perform better. Overall I feel the paper helps understanding how neural network works in practice. It can be better if an easy to understand intuition of why overparameterization helps (both proof and practical problems) is provided. ==== The authors have answered my concern on better explaining the intuition. I'll keep my score.

Reviewer 3



This work studies optimization of $\ell_2$-loss for training multilayer nonlinear recurrent neural network. The paper is well-written, well-presented, and easy to follow. When each layer of the neural network is highly over-parameterized, with several additional assumptions, it proves that the (stochastic) gradient descents converges linearly to zero training loss. The overall theoretical result is impressive, and several new results have been developed recently based on this work. The major concern is the practicality and generality of the result for real applications. Below are some more detailed comments: 1. However, the major concern is the practicality of the assumptions and hence the results. The theory seems to suggest that random initialization is already good’’ enough, that one only needs to make small adjustment of the weights to obtain zero training loss. Recent work [L] Lenaic Chizat, Edouard Oyallon, Francis Bach, On Lazy Training in Differentiable Programming’’, 2018. [G] Gilad Yehudai and Ohad Shamir. On the power and limitations of random features for understanding neural networks. arXiv preprint arXiv:1904.00687, 2019. explains that the regime (This is a regime where small changes of the weights result in large change of function values.) of the model the authors considered in this submission tends to behave like linear models (where they call lazy training), which seems to be not yet sufficient to explain the success of deep neural network in practice (demonstrated by experimental results on deep CNN). The authors should at least cite these results, and provide a detailed discussion on this. 2. The original idea of this line of work is coming from the neuron tangent kernel. This is from infinite width to finite width (asymptotic to non-asymptotic). The authors need to cite and recognize their work Arthur Jacot, Franck Gabriel, Clément Hongler. Neural Tangent Kernel: Convergence and Generalization in Neural Networks. 3. It seems quite strange to the reviewer that the matrix A and B can be set to their random initialization without optimization, and the training loss can be optimized to 0. Under the considered setting here, it seems that random initializations are close to the optimal solution. In the result, it seems that gradient descent on the weights W (given the stepsize eta very small in the theory) makes very little adjustment (the properties in Theorem 3 and Theorem 4 only hold when W is very close to the initialization), but producing zero training loss. 4. In the main theorem, the authors hide the degree of polynomial of the overparameterization in m. The degree of the polynomial seems to be very high. Can the author give us a sense how loose are the bounds, and what is the conjecture dependence for the result to hold here? 5. References. It would be great if the authors can make the citations in order of appearance and abbreviate them (e.g., page 1, change [58,19] to [1-2], and [7,49,54,31,15,18,36,60,59] should be abbreviated [3-11], etc). ====== After Rebuttal ======= The rebuttal clarifies and addresses most of my concerns, especially the relation of this work to [L], [G], and [J]. Albeit its practicality (it requires significant overparameterization), I agree with other reviewers that this is a solid theory paper that has triggered a lot interest in recent theoretical understandings of deep neural networks (e.g., SGD on the training loss of multilayer ReLU network). It also provides some explanation how SGD works in the early stages of training neural network.

[Author Response · NeurIPS 2019]

We thank all the reviewers for the time reading our paper! We especially thank **R4** for acknowledging that this is "a
strong theory paper" and "the techniques developed can be very useful in the analysis of multi-layer ReLU nets."

We also thank **R6** for appreciating the presentation of this paper and shall continue to improve the writing to hopefully
make **R5** satisfied as well. We will fix all the minor issues, and below we only address the main concerns.

- 5 **R4+R6** has concerns about the polynomial network size not being applicable in practice
Indeed, in practice, the step size is usually larger so we expect a different behavior. In this work, we give the first
result proving that in the idealized setting, namely "large over-parameterization and small learning rate", neural
network as complicated as RNNs can be trained to zero training error.

We believe this can at least provide intuitions on the second phase of NN learning: where one decays the learning
rate and the training error can go to zero. Our $m$ dependency is the worst-case theoretical bound that holds for
all possible inputs and labels. In practice, one usually expects a more benign training set and the bounds can be
improved by a lot. (This is indeed the case when data is generated from some low-complexity concept class, where
$m$ no longer depends on a big polynomial over input size $n$, see follow up [1].)

- 14 **R5:** Can the authors provide an easy-to-understand paragraph describing why GD/SGD will reach global minimum
and how overparameterization helps?
We will try. What we have provided in the current version is a 6-paged sketched proof (Sections 4+5 on pages 6-8
and Sections 7+8 on pages 13-16) so that the readers don't need to go to our appendix and can already understand
our proofs and why over-parameterization helps. We're glad to see that Reviewer 6 has liked this structure. We can
try to provide a half-page sketch for this 6 pages sketched. **Do you think that will help?**

- 20 **R6** has pointed out 3 papers and asked us to compare with: [L] "On Lazy training..." arXiv 1812.07956, [G] "On
the power ..." arXiv 1904.00687, and [J] Jacot et al. "Neural Tangent Kernel..."
Note that [L] and [G] both appeared *after* our work (we appeared in October 2018), but we can cite all of them.

In particular, [J] studies the infinite-width setting of neural network, which is worse than our *polynomial width*
result. It's our fault to forget to cite it.

As for the criticism raised by [L] and [G] about lazy training, here's our response. "Lazy training" has several
magnitudes. Perhaps the "laziest" is when NN can be approximated by a linear model (like this paper). The less
lazy one is to take into account interactions between layers (see follow up [2]), and the least lazy one is to take into
account also sign changes (see follow up 1905.10337). As it goes less lazy, the power of NN becomes stronger.
Perhaps surprisingly, in this laziest model, we can already prove that (R)NN memorizes data. (This is non-trivial!
Since (1) why is RNN close to linear model in small learning rate regime? and (2) even so, why can linear model
train to zero error?)

Finally, we believe studying this laziest model is very meaningful. Not only it can give us intuitions about the
second phase of NN learning (where one decays the learning rate), it also gives us technical tools for studying less
lazy models (such as follow ups [2] and 1905.10337, both relying on this paper). Most significantly, the perturbation
theorems proved in this paper is at the heart of all of those follow-ups.

- 36 **R6:** it seems strange $A, B$ can be set random, $W$ moves little, and the network can produce zero training loss.
There's no contradiction here. Recall when $A, B, W$ are all at random initialization, (we have proved) the RNN
outputs are of magnitude roughly constant. We claim by moving from $W$ to $W + W'$, even when $\|W'\|_2 \ll \|W\|_2$,
we can already change the output significantly (i.e., by more than a constant). There is no contradiction here:

   - 40 $W$ is random, so when interacting with $A$ and $B$, there is a lot of cancellation and the output is small;
   - 41 $W'$ correlates with $A, B$, so when interacting with them, gives a much bigger output.

Hence, $W'$ need not be as big as $W$. As a toy example, when $a, w \in \mathbb{R}^m$ are random vectors with coordinates
i.i.d. from $\mathcal{N}(0,1)$, then $|\langle a, w \rangle| \approx \sqrt{m}$ but $\|a\|_2 \approx \|w\|_2 \approx \sqrt{m}$. If we update $w + w' = w + \frac{1000}{\sqrt{m}} a$, then
$\langle a, w + w' \rangle \approx 1000\sqrt{m}$ which is very different from $\langle a, w \rangle$ but $\|w'\| \ll \|w\|$.

(There is practical evidence of this as well. One example is for 2-layer network on MNIST, see Fig. 5 on page 26 of
prior work Li-Liang [30] of their NeurIPS camera ready version. There are also evidence in some follow-up works
of the anonymous authors. We can consider adding some explanations of this in a next revision.)

[Meta-Review · NeurIPS 2019]

This paper proves poly-time convergence of SGD/GD in over-parametrized RNNs for the first time. Given that there is not many theoretical results in this space. All reviewers find this result a significant progress. Therefore, I recommend acceptance.